# The Role of Microglia during West Nile Virus Infection of the Central Nervous System

**DOI:** 10.3390/vaccines8030485

**Published:** 2020-08-28

**Authors:** Sarah Stonedahl, Penny Clarke, Kenneth L. Tyler

**Affiliations:** 1Department of Immunology and Microbiology University of Colorado Anschutz Medical Campus, Aurora, CO 80045, USA; sarah.stonedahl@cuanschutz.edu; 2Department of Neurology, University of Colorado Anschutz Medical Campus, Aurora, CO 80045, USA; 3Department of Immunology and Microbiology, Infectious Disease, Medicine and Neurology, University of Colorado Anschutz Medical Campus, Aurora, CO 80045, USA; 4Department of Veterans Affairs, Aurora, CO 80045, USA

**Keywords:** West Nile Virus, microglia, central nervous system, neuroinflammation

## Abstract

Encephalitis resulting from viral infections is a major cause of hospitalization and death worldwide. West Nile Virus (WNV) is a substantial health concern as it is one of the leading causes of viral encephalitis in the United States today. WNV infiltrates the central nervous system (CNS), where it directly infects neurons and induces neuronal cell death, in part, via activation of caspase 3-mediated apoptosis. WNV infection also induces neuroinflammation characterized by activation of innate immune cells, including microglia and astrocytes, production of inflammatory cytokines, breakdown of the blood-brain barrier, and infiltration of peripheral leukocytes. Microglia are the resident immune cells of the brain and monitor the CNS for signs of injury or pathogens. Following infection with WNV, microglia exhibit a change in morphology consistent with activation and are associated with increased expression of proinflammatory cytokines. Recent research has focused on deciphering the role of microglia during WNV encephalitis. Microglia play a protective role during infections by limiting viral growth and reducing mortality in mice. However, it also appears that activated microglia are triggered by T cells to mediate synaptic elimination at late times during infection, which may contribute to long-term neurological deficits following a neuroinvasive WNV infection. This review will discuss the important role of microglia in the pathogenesis of a neuroinvasive WNV infection. Knowledge of the precise role of microglia during a WNV infection may lead to a greater ability to treat and manage WNV encephalitis.

## 1. Introduction

West Nile Virus (WNV) is a neurotropic, mosquito-borne, single-stranded RNA flavivirus maintained in an enzootic cycle between mosquitos, such as *Culex pipiens*, and perching birds (order Passeriformes), such as crows, jays, and finches [1,2]. While birds are the natural reservoir for WNV, an infected mosquito may bite other mammals such as horses and humans and transmit the virus through saliva [2]. After an infected mosquito bites a human, the virus replicates within dendritic cells and macrophages of local tissue near the site of inoculation and subsequently spreads to regional lymph nodes. The resulting viremia then spreads the virus throughout the body [2]. West Nile Virus viremia in humans is not high enough or sustained enough to support subsequent transmission to mosquitoes, thus humans are designated “dead-end” hosts [1,2].

West Nile Virus infection is now the most common cause of epidemic viral encephalitis in the United States. According to the Center for Disease Control (CDC), since its introduction into North America in 1999, WNV has caused over 50,000 confirmed cases of disease and almost 24,000 cases of neuroinvasive infection [1,3,4]. The number of total WNV cases is likely an underestimate as mild and asymptomatic cases are not generally reported to the CDC. Of note, WNV epidemics in 2003 and 2012 were the largest outbreaks of neuroinvasive viral infections ever reported in the Western Hemisphere [5,6]. About 80% of human WNV infections are asymptomatic and 20% of infections result in an acute illness, known as West Nile Fever, which is characterized by fever, headache, fatigue, anorexia, nausea, myalgia, and lymphadenopathy [2]. Less than 1% of human WNV infections lead to severe neurological disease including meningitis (inflammation of the membranes covering the brain and spinal cord), encephalitis (inflammation of the brain), or acute flaccid paralysis (rapid onset of weakness and loss of muscle tone) [2]. Mortality in patients with neuroinvasive disease is reported at around 10%, and 50% of surviving patients have long lasting neurological sequelae [2,3,4,7].

While effective equine vaccines against WNV have been licensed for a long time, no human vaccine is currently available. However, a variety of vaccine candidates have been developed and evaluated to different degrees in preclinical trials including those based on inactivated viruses, recombinant and chimeric viruses, purified viral proteins, and other approaches [8,9,10,11,12,13,14,15]. As there are currently no available human vaccines or antivirals targeting WNV, preventative methods such as the use of mosquito nets and control of mosquito populations are the predominant strategy for managing WNV infections in humans [1].

## 2. West Nile Virus-Induced CNS Disease

West Nile Virus-induced CNS disease occurs as a result of virus crossing the blood brain barrier (BBB) following systemic infection. Mechanisms of viral entry into the CNS are not yet fully understood. However, theories include a cytokine-mediated increase in permeability of the BBB, hematogenous entry by infection of infiltrating immune cells, and retrograde axonal transport [7,16,17]. The greatest risk factor for developing WNV neuroinvasive disease is older age (>64 years), while other risk factors include hypertension and diabetes [18]. Certain genetic factors also dictate increased susceptibility to neuroinvasive disease including single nucleotide polymorphisms in the oligoadenylate synthetase (OAS) gene (involved in antiviral innate immunity) and genetic deficiencies in C-C chemokine receptor type 5 (CCR5), which inhibit lymphocyte trafficking into the CNS [19,20].

Once within the CNS, WNV directly infects neurons and several studies demonstrate that caspase 3-dependent apoptosis is an important mechanism of neuronal cell death and CNS injury during WNV infection [21,22]. The pathology of neuroinvasive WNV infection likely results from both apoptosis of neurons and secondary effects of the neuroinflammatory response, including damage to bystander cells and gliosis [22]. A critical component of the neuroinflammatory response induced by WNV infection is the activation of microglia, the innate immune cells of the CNS [23]. In this review, we will discuss the role of microglia in WNV-induced CNS disease as a contributor both to the protective innate immune response and to the development of long-term neurological damage.

## 3. Microglia Become Activated during WNV Infection

Microglia are of myeloid origin and arise from the yolk sac during early development [24,25,26]. These cells are the resident mononuclear phagocytic cells of the CNS and play multiple roles in the CNS such as immune defense, maintenance of homeostasis, and synaptic pruning during early development [27,28,29]. Importantly they continually monitor the CNS for signals of injury and infection [28,30,31]. Microglia express a spectrum of activation states in response to environmental and pathogenic cues [32,33]. Morphological changes of activated microglia include a shift from a ramified state, characterized by long extending processes, to a larger more amoeboid phenotype [34]. In addition, activated microglia display increased motility, proliferation and production of inflammatory cytokines and chemokines [29,35,36,37]. In neuroinflammatory and neurodegenerative conditions, microglia are thought to contribute to pathogenicity [36,38,39], however, in the case of viral encephalitis, it remains unclear whether microglia are beneficial or detrimental to disease outcomes, or possibly both. 

Classically, macrophages and microglia have been defined as being polarized, meaning they express two major activation states: pro-inflammatory M1 and anti-inflammatory M2 [40]. While some individuals feel that this is not an accurate description of microglial activation [32], it remains important to consider the role of microglia polarization in the pathogenesis of WNV infections. The ability to shift phenotypes allows microglia to maintain homeostasis within the CNS. Following infection of the CNS, microglia express markers characteristic of M1 activation including increased expression of Iba-1, as well as chemokines such as C-C motif cytokine 2 (CCL2), CCL3, CCL5, and CCL7 [41]. The anti-inflammatory drug minocycline reduces the expression of M1 inflammatory genes and increases the expression of M2 genes, following WNV infection of CNS tissue [42]. This shift of microglia towards a more anti-inflammatory M2 state was neuroprotective [42] and suggests that M1 microglia are required for initial control of WNV infection; however, switching to an M2 phenotype may be critical in preventing excessive damage due to inflammatory processes. For the purpose of this review, we will mostly be referring to M1 microglial polarization when discussing activation of microglia. 

Several neurotropic viruses have been shown to induce microglia activation in the CNS following infection, including Japanese encephalitis virus [43,44], Dengue Virus [45], Zika virus [46], Tick-borne encephalitis virus [47], reovirus [48], and mouse hepatitis virus [49,50]. Evidence that microglia are activated during WNV-induced CNS disease in humans includes the presence of microglial nodules and increased proliferation [51,52]. Activated microglia are also seen in the CNS of mice infected with WNV [53,54,55] and in WNV-infected ex vivo brain and spinal cord slice cultures [21,41]. The use of ex vivo slice cultures allows for isolation of the CNS immune system from the peripheral immune response. In this method, spinal cords or brains are harvested from neonatal mice and used to create 300-µm-thick slice cultures, which are incubated on culture inserts prior to infection. Using this model, activated microglia were identified following WNV infection by increased expression of ionized calcium binding adaptor molecule 1 (Iba1) [21,41], a microglia-specific marker [56]. In addition, microglia in WNV-infected ex vivo slices were increased in both size and number compared to uninfected slices and displayed an amoeboid phenotype [21,41]. Distinct microglia cell processes related to cell motility and phagocytosis of WNV-infected cells and antigenic debris were seen in WNV-infected ex vivo slice cultures (Figure 1), including microglial cellular projections stretching over a range of distances to reach WNV-infected cells, which were often filopodial/lamellipodial in appearance (Figure 1A), reflecting microglia with amoeboid morphology [41]. These microglia cell processes were observed contacting infected cells and initiating engulfment activity (Figure 1B–E). The formation of phagosomes was also prevalent (Figure 1F). Notably, microglia virtually never appeared to be infected themselves, despite taking in material (cells and debris) that stained positive for WNV antigens.

Intercellular communication via cytokine/chemokine signaling drives inflammatory responses during viral infections of the CNS [40,57]. Infection of CNS tissue with WNV in vivo [21,58] and in ex vivo [41] and in vitro [23] models of WNV pathogenesis results in increased expression of proinflammatory cytokines and chemokines associated with microglial activation including C-X-C motif chemokine 10 (CXCL10), CXCL1, CCL5, CCL3, CCL2, tumor necrosis factor alpha (TNF-α), TNF-related apoptosis-inducing ligand (TRAIL), and interleukin-6 (IL-6) [40]. These cytokines are involved in inflammatory processes and activating/recruiting immune cells such as T cells [59]. Inhibition of these chemokines/cytokines following treatment with minocycline resulted in decreased neuronal cell death following WNV infection of ex vivo spinal cord slice cultures [42], suggesting they contribute to disease; however, depletion of microglia did not significantly change chemokine/cytokine expression in the brains of WNV-infected mice [60] so it is unclear what role microglia play in the production of these proteins.

Based on the evidence presented above, it can be concluded that microglia become activated following infection of the CNS with WNV. These activated microglia can initiate an immune response that is intrinsic to the CNS and does not require input from the peripheral immune system as seen in the ex vivo slice culture model. Activation of the microglia is characterized by morphological changes and is associated with the increased production of inflammatory cytokines/chemokines.

## 4. Microglia Recognize and Respond to WNV through a Variety of Receptors

Toll-Like Receptors (TLRs) are membrane spanning receptors commonly expressed by cells such as macrophages, dendritic cells, and microglia, which monitor the body for signs of infection. They play key roles in the innate immune system [61] and have the ability to recognize structurally conserved pathogen-associated molecular patterns (PAMPS) present on pathogens [61,62,63]. Microglial expression of TLR3, which recognizes double-stranded viral RNA, is thought to play an important role in the detection of viruses, including WNV, in the CNS [64]. 

Several studies have shown that TLR3 signaling plays a protective role in preventing lethal WNV encephalitis [54,65,66]. For example, mice deficient in the expression of TLR3 [63,64] and Myeloid Differentiation Primary Response 88 (MyD88), which functions downstream of TLR3 [54], show increased susceptibility to WNV that is associated with higher viral loads in the brain and increased expression of inflammatory cytokines. While these studies present evidence for a protective effect of TLR3 following WNV infection, an earlier study showed increased survival from WNV infections in TLR3-deficient mice [65]. In this study, TLR3-deficient mice were more resistant to lethal WNV infection and showed impaired cytokine production along with enhanced viral load in the periphery. However, in the brain, viral load, inflammatory responses and neuropathology were reduced compared to wild-type, indicating TLR3 may mediate entry of WNV into the brain [67]. It has been suggested that the difference in outcome seen in TLR3-deficient mice could be explained by the distinct route of inoculation (subcutaneous versus intraperitoneal), passage history of the virus (mammalian Vero cell versus insect-cell derived), and/or the virus dose [63]. More research will need to be done to achieve a full understanding of the role of TLR3 signaling in WNV encephalitis.

Retinoic Acid Inducible Gene I (RIG-I) and Melanoma Differentiation Antigen 5 (MDA5) are cytosolic pattern recognition receptors (PRRs), which recognize viral RNA products, including WNV genetic material, and are critical for initiating an innate immune defense [68,69]. Lack of either RIG-I or MDA5 in mice results in decreased innate immune signaling and control of viral growth. A double knockout of both receptors leads to complete loss of innate immunity gene function in WNV-infected cells and severe disease in mice [68]. Microglia constitutively express both RIG-I and MDA5 [70]. The downstream central adaptor molecule mitochondrial antiviral-signaling protein (MAVS) is also essential for control of WNV infections. A deficiency in MAVS in hematopoietic cells, such as microglia and macrophages, leads to increased mortality in mice infected with WNV [71]. These findings indicate the importance of the MAVS signaling pathway in the microglial response to WNV particles within the CNS. 

## 5. Microglia Are Critical for Protection from WNV Encephalitis

As documented above, microglia recognize and become activated in response to WNV infection. However, it remains unclear if this process is essential to effective host protection against WNV or whether activated microglia contribute to pathogenesis. For example, following infection with WNV, both decreased [53] and increased [54,64] microglial activation have been associated with increased neuronal death and susceptibility to disease. To address this disparity, the effect of WNV infection on mice lacking microglia has been investigated. Mice deficient in the expression of interleukin 34 (IL-34), which lack microglia but retain bone marrow-derived macrophages, have increased susceptibility to WNV, suggesting a role for microglia in antiviral defense [72]. In addition, recent studies performed in mice in which microglia were depleted using Plexxicon 5622 (PLX5622), an inhibitor of colony-stimulating factor 1 receptor (CSF1R), demonstrated a significant increase in susceptibility to WNV-induced CNS disease [60]. Colony-stimulating factor 1 is critical for microglia survival [72,73,74], and PLX5622 has been shown to have specificity towards microglia without affecting other cell populations such as brain-specific macrophages, peripheral macrophages, or lymphocytes [49,75,76]. Microglial depletion using PLX5622 does not alter brain size, cognition, or motor function in mice [76]. 

To deplete microglia, mice were fed a diet containing PLX5622 for 10 days resulting in greater than 80% depletion of their microglia population throughout the brain when compared to mice fed a control diet [60]. Following infection of PLX5622-treated mice with WNV, brain titers were significantly (10–100-fold) higher than in control mice at days 6, 9, and 10 post infection. In addition, mortality dramatically increased. Of the mice infected with WNV, 100% of PLX-treated microglia-depleted mice died compared to only 25% of the control fed mice [60]. These studies indicate that microglia play a critical role in controlling WNV infection and limiting WNV-induced mortality. 

Results in WNV-infected PLX5622-treated mice were remarkably similar to another recent study, which found that microglia are crucial for protection of mice from neurotropic coronavirus encephalitis and may be required for the generation of an efficient T cell response [49]. This study also showed that microglial protection was time dependent [49]. Elimination of microglia using PLX5622 between days 0 and 6 post-infection resulted in increased mortality in infected mice; however, microglial depletion after the first 6 days showed no effect on mortality [49]. This data indicates that microglia are essential for limiting viral growth and neuronal cell death during the initial phase of a neuroinvasive viral infection but become less important during later infection, when adaptive immune responses have developed. Determination of whether microglia follow a similar time-dependent role following WNV infection in the CNS and whether this is associated with changes in the T cell response remains to be determined. 

Collectively, the studies discussed above reveal the critical role microglia play in protecting the host from WNV-induced CNS disease. Elimination of microglia during WNV encephalitis led to increased viral titers within the brain and increased mortality in mice. Further studies are needed to determine the mechanism by which microglia protect the CNS from WNV infection. Potential mechanisms include phagocytosis of WNV-infected neurons, production of inflammatory cytokines/chemokines, and signaling to the adaptive immune system. 

## 6. Microglia May Contribute to Entry of WNV into the Brain

Matrix metalloproteinases (MMPs) are a family of proteins whose major function is remodeling the extracellular complex (EC) of the BBB [77,78]. They have also been shown to play a role in immunity by modulating the trafficking of immune cells into the CNS [79]. MMP9, is expressed by microglia [78], and has been shown to be important for WNV entry into the brain [77]. MMP9 knockout mice infected with WNV show decreased viral loads in the brain, but not the periphery, and increased survival when compared to wild-type (WT) mice [77]. Resistance to lethal WNV infection coincided with an intact BBB as shown by reduced Evan’s Blue Dye leakage into the brain, indicating a role for MM9P in altering permeability of the BBB during a WNV infection [77]. 

An additional mechanism by which microglia may contribute to WNV entry into the brain involves intercellular adhesion molecule (ICAM-1), a surface glycoprotein involved in cellular extravasation into sites of inflammation. ICAM-1 is expressed by several cell types including microglia [80]. The expression of ICAM-1 on microglia may contribute to extravasation of leukocytes via the “Trojan horse” method, in which the virus gains access to the CNS through infected leukocytes [17,81]. Indeed, WNV infection has been shown to induce ICAM-1 expression in human endothelial cells and mouse brains, leading to disruption of the BBB and increased leukocyte infiltration [82]. It was also found that this process can be reversed through the use of blocking antibodies against ICAM-1 [82]. 

## 7. T Cells Promote Microglia-Mediated Loss of Synapses Following WNV Infection

Individuals recovering from WNV encephalitis may experience long-term neurological deficits resulting from the infection [3]. This effect of WNV infection of the CNS is being extensively studied as an increased understanding of why these impairments occur will likely lead to an improvement in the long-term outcomes of recovering patients. A predominant theory is that microglia activated by memory T cells may contribute to neuronal damage following infection with WNV [83,84]. Following initiation of innate immune responses, activated microglia release cytokines that signal the peripheral immune cells to infiltrate the CNS to clear the viral infection [24,28]. CD8^+^ T cells mediate viral control through the secretion of effector molecules such as granzymes, perforin, and interferon-γ (IFNγ) [85]. While CD8^+^ T cells are necessary for proper clearance of virus from the CNS [86,87], they also seem to contribute to neurological deficits following WNV infection [83,84,88]. Once in the CNS, CD8^+^ T cells will differentiate into brain resident memory T cells, which are important for protection from reinfection [85]. These T cells remain within the CNS for months to years following neurotropic viral infections and studies have shown that these memory T cells promote microglia-mediated synaptic elimination following neuroinvasive WNV infections contributing to cognitive impairment. [83,88,89]. 

A recent study used a mouse model of viral encephalitis to show that T cells promote microglia-mediated loss of synapses following infection with WNV [83]. This study found an increase in the number of microglia-expressing markers indicative of activation induced by the T cell cytokine IFN-γ [83]. Analysis by qPCR also revealed increased expression IFN-γ in the hippocampus following infection and throughout recovery. In addition, mice deficient in the expression of IFN-γ and WT mice were infected with WNV. Mice lacking IFN-γ showed no differences in survival or weight loss when compared to WT mice [83]. However, WT mice showed special learning deficits on the Barnes maze test, while *Ifngr1-/-* mice were protected from this outcome [83]. This study shows that during the process of viral clearance following WNV infection, IFN-γ produced by memory T cells can alter the function of microglia leading to elimination of synapses. 

The complement cascade is a component of the innate immune response, which plays a key role in defense from pathogens. During early development, microglia are involved in the process of synaptic pruning within the nervous system, which is dependent on the classical complement cascade [90,91,92,93]. The mechanism by which the IFN-γ activated microglia eliminate synapses seems to be dependent on the complement system. It was found that brain regions retrieved from mice that had recovered from WNV showed increased expression of genes involved in microglial synaptic elimination by components of the complement system such as C1QA [84]. Mice with either fewer microglia (due to lacking the CSF1R ligand IL-34) or a deficiency in the C3 complement receptor showed increased protection from microglia-mediated synapse elimination [84]. This study suggests that the complement cascade and especially microglial-C3 is responsible for synaptic elimination seen in mice following WNV infection in the brain. Collectively, the studies discussed in this section provide evidence for a mechanism of neuronal damage following neuroinvasive WNV infection involving interactions in which microglia are promoted by memory T cells to eliminate synapses that have been targeted by the complement cascade. Understanding the dynamics between T cells and microglia following WNV infection of the CNS and how microglia may contribute to neurological sequalae could provide important insight into potential treatments for patients recovering from WNV encephalitis. 

## 8. Conclusions

Microglia maintain a variety of functions within the CNS, which are critical for protection from pathogens, homeostatic maintenance, and early development of the CNS. These roles are summarized in Table 1. In this review, we have highlighted the important role microglia play during a neuroinvasive WNV infection. Although it is evident that microglia are essential for protection from WNV, the mechanism behind how microglia limit viral growth and mortality remains unclear. The multiple and complex roles of microglia during WNV infections are pictured in Figure 2. Future studies should focus on the precise mechanisms by which microglia protect from WNV and enhancing the understanding of neuroinflammatory processes that accompany neuroinvasive WNV infections. It still remains to be determined which cytokines/chemokines are produced by microglia during a WNV infection and why a decrease in the expression of inflammatory cytokines/chemokines in not observed in the CNS of mice following WNV infection in the presence of PLX5622. 

While microglia are protective during the initial infection and vital for resolution of the infection, potentially dysregulated microglia responses driven by T cells may contribute to neurological damage and long-term impairment during the recovery stages. Targeting microglial responses during both the initial and late stages of infections may offer new therapeutic options for human patients with neuroinvasive WNV infections. For example, it may be beneficial to enhance the activity of microglia during the initial infection to provide enhanced protection but then limit microglial activation during the recovery stages of infection to improve long-term outcomes. One promising treatment that may be used during the initial WNV infection to improve survival is the use of granulocyte-macrophage colony-stimulating factor (GM-CSF), which promotes growth and activation of microglia, macrophages, and other myeloid cells [75,94]. This drug is currently available (Leukine) and is used to treat leucopenic patients as well as being investigated as a treatment for Alzheimer’s disease [95]. Another study showed that the anti-inflammatory drug minocycline may be beneficial in recovery from a WNV infection of the CNS [42]. 

Most of the studies presented in this review utilized a knockout mouse model or mouse tissue model to investigate microglia. However, it is important to understand the limitations of using mouse models when studying microglia and WNV pathology. Human microglia exhibit several differences to mouse microglia including increased levels of several immune genes and being longer-lived, which may lead to altered outcomes when comparing human and mouse infections [96]. It is vital to the complete understanding of the role of microglia during a WNV infection to use mouse studies as a compliment to human studies and not a replacement. Understanding exactly how microglia influence the progression and recovery of WNV neuroinvasive infections will allow for the development of specific treatments, which could provide the optimal outcome to patients recovering from this infection. 

## Figures and Tables

**Figure 1 vaccines-08-00485-f001:**
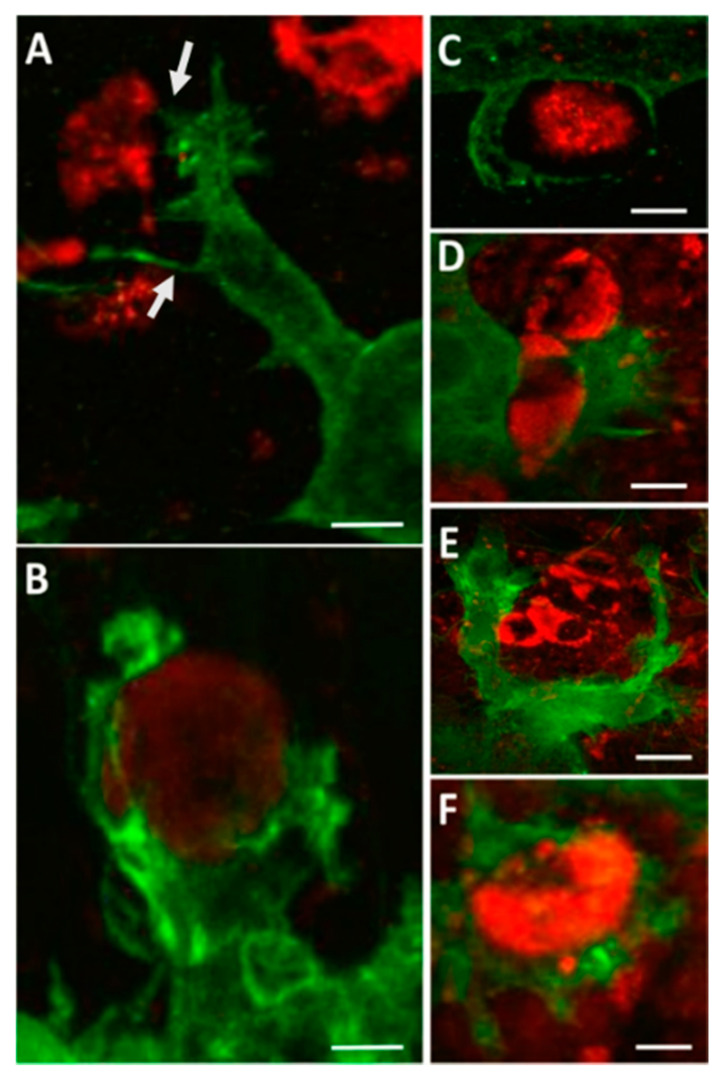
Microglia phagocytic processes in WNV-infected CNS tissue. WNV-infected spinal cord slice cultures (SCSC) samples were collected at 6 days post infection (dpi) and processed for immunohistochemical staining with Iba1 (green) and WNV-E (red). (**A**) Fine filopodial projections directed toward WNV antigenic material are observed protruding from a lamellipodial projection of a microglia cell. Bar, 12 μm. (**B**) Microglial processes begin to engulf a WNV-infected cell, depicting the formation of a structure known as a phagocytic cup. Bar, 6 μm. (**C**) Microglia processes surround WNV antigenic material. Bar, 5 μm. (**D**) A microglia cell in the process of engulfing a WNV-infected cell. Bar, 9 μm. (**E**) Large lamellipodial processes from a single microglia cell surround a cluster of WNV-infected cells. Bar, 20 μm. (**F**) A WNV-infected cell is completely engulfed by a microglia cell, depicting the formation of a structure known as a phagosome. Bar, 8 μm. Quick et al. 2014 [41].

**Figure 2 vaccines-08-00485-f002:**
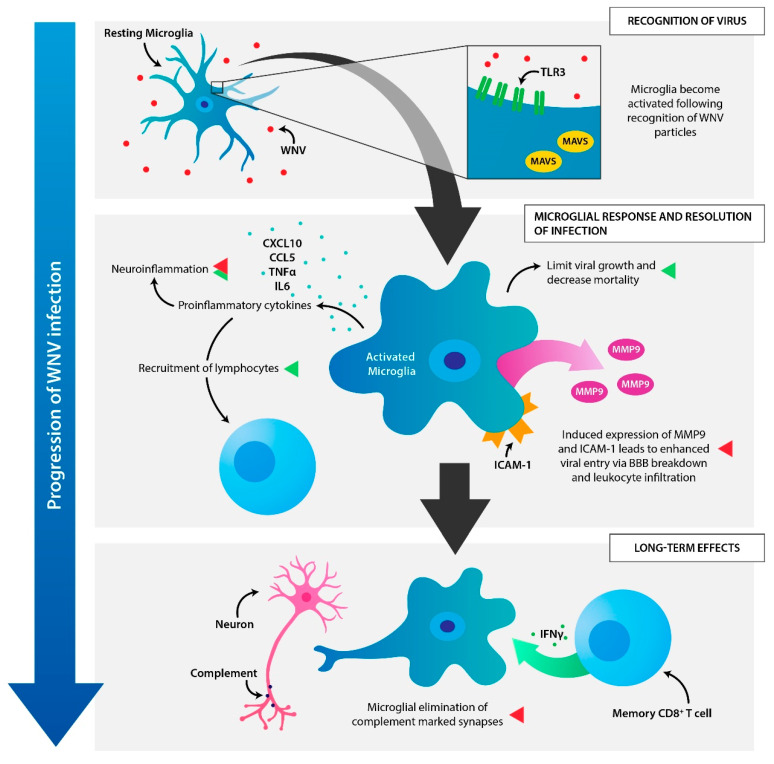
The many roles of microglia during a WNV infection. RECOGNITION OF VIRUS: Microglia recognize WNV viral particles in the CNS through multiple signaling pathways, including TLR3/MAVS [1,2]. MICROGLIAL RESPONSES AND RESOLUTION OF INFECTION: Microglia become activated in response to WNV infection and initiate morphological and functional changes [3]. Activated microglia control viral growth and reduce mortality in mice infected with WNV [4], likely due to a combination of microglial phagocytosis of infected neurons and the release of inflammatory cytokines, which contribute to neuroinflammation and the recruitment of lymphocytes [3,5]. In some cases, released cytokines may contribute to WNV-induced pathogenesis. MMP9 and ICAM-1 expression by microglia may enhance viral entry into the CNS through breakdown of the BBB and extravasation of potentially infected leukocytes [6,7]. LONG TERM EFFECTS: Following infiltration of T cells into the CNS, microglia may contribute to synaptic damage triggered by IFNγ produced by memory CD8^+^ T cells and enhanced by the complement cascade [8,9]. This synaptic elimination leads to neurological damage during recovery from WNV encephalitis [9]. Green and Red indicator arrows designate protective (green) and non-protective (red) microglial responses.

**Table 1 vaccines-08-00485-t001:** Role of microglia in the CNS before, during, and after neuroinvasive WNV infection.

Microglial Role	Stage	Mechanism	Result of Microglial Action	Reference(s)
Synaptic pruning	Embryonic development	Compliment system	Proper development of the CNS	[90,91,92]
Active monitoring of the CNS for signs of pathogens or damage	Prior to infection	TLRs, PRRs	Activation of microglia in response to pathogenic markers	[28,30,31]
Entry of WNV into the CNS	During infection	Expression of MMP9 and ICAM-1	Enhanced viral entry into the CNS through breakdown of the BBB and infiltration of leukocytes	[77,82]
Recognition of WNV	During infection	TLR3, RigI, Mda5, MAVS	Activation of microglia	[64,65,68,69,71]
Limit viral growth	During infection	Unknown	Decrease viral load in the CNS	[60]
Production of inflammatory cytokines	During infection	Intracellular signaling cascades triggered by extracellular receptors such as TLRs	Modulation of neuroinflammation	[21,41,58]
Recruitment of T Cells	During infection	Cytokines	Infiltration of T cells into the CNS and localization to the site of infection	[23]
Compliment-mediated synaptic elimination	Recovery	CD8^+^ T cell production of IFNγ and the compliment system	Long term neurological damage	[83,84]

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
