# Peer review of "The Role of Microglia during West Nile Virus Infection of the Central Nervous System"

_vaccines, 2020, doi:10.3390/vaccines8030485_

Round 1
Reviewer 1 Report
The review article by Stonedahl et al entitled “The role of microglia during West Nile virus (WNV) infection of the central nervous system” discussed about the role of microglia in regulating WNV mediated neuroinflammation. The paper is very well written and provide a substantial summary about the importance of microglia in WNV infection.
There are few concerns that can be included to improve the quality of this paper.
- Need some discussion about M1 and M2 microglia and their role in WNV pathogenesis.
- Majority of microglial studies has been performed using knockout mice model. An earlier interesting study had shown that human microglial cells express increased level of several immune genes like TLR, FCY, SIGLEC, TAL1 and IFI16 compared to mice microglia (PMID 2671693). The authors should discuss the limitation of the WNV research in mice model as the aging of microglia in human might a different impact on WNV pathogenesis compared to mice.
- MMP9 is also expressed in microglia and its role in WNV pathogenesis can be cited (PMID 1632868).
- ICAM-1, expressed in microglia and their role in WNV neuro-invasion needs to be discussed.
- Role of complement C3 receptor and IL34 producing microglia in WNV infection should be included (PMID 27337340).
- Table 1: Some of the microglia roles cited in the table are not from WNV study. For example, references 40, 41 and 37 are not related to direct WNV study.
Reviewer 2 Report
The manuscript addresses the important role of microglia in the pathogenesis of west Nile Virus infection. In terms of manuscript strengths, the subject is great, and the manuscript could be read and understood with ease. The manuscript is generally sound, and the studies reviewed are convincing and of interest to the field. I believe the manuscript will be strengthened by some careful rewriting. Here I have raised a few points for the authors to revise the manuscript and hopefully improve the quality of the manuscript:
Major points:
- The review accomplished a comprehensive view of the research area, but the manuscript lacks an original figure that summarizes the revised role. I believe that the addition of figures will improve the reader's understanding and the quality of the manuscript. I strongly recommend that a figure is included, to illustrate the role of microglia in the CNS after neuroinvasive WNV infection.
- I have noticed that figure 1 was published by the same research group of the authors of this manuscript. Did all members of the previous work agree to publish this figure? Did the previous scientific journal also authorize this publication? Please, I ask the authors to pay special attention to this point.
Minor points:
- The entire manuscript must be reviewed for identification and use of acronyms. (E.g. DPI - figure 1 subtitle; WT- page 6 line 249 ; Acronyms used without identification in the first mention in the text).
- Page 1 line 36 “such as “Culex pipiens”. Please, highlight the species name in italics
- Page 1 line 36 change “perching birds” for “order Passeriformes”
- Page 1 line 37 Please add “crows” in “…as jays and finches…”.
- Some phrases with important information are without reference. Please add the appropriate references. Below I identified the page and line corresponding to the end of the sentence:
- Page 1 lines 37; 43
- Page 2 lines 52; 55; 81; 89
- Page 5 lines 214; 244; 245; 246; 248
- Figure 1: I suggest adding arrows to facilitate the visualization of the structures described in subtitles.
- Page- 3 lines 98-100. Authors should consider adding the following works with other flaviviruses to this part of the manuscript.
- Carola Maffioli, PhD, Denis Grandgirard, PhD, Olivier Engler, PhD, Stephen L. Leib, MD, A Tick-Borne Encephalitis Model in Infant Rats Infected With Langat Virus, Journal of Neuropathology & Experimental Neurology, Volume 73, Issue 12, December 2014, Pages 1107–1115, https://doi.org/10.1097/NEN.0000000000000131
- Jhan, Ming-Kai et al. “Dengue virus infection increases microglial cell migration.” Scientific reports vol. 7,1 91. 7 Mar. 2017, doi:10.1038/s41598-017-00182-z
- Xu P, Shan C, Dunn TJ, Xie X, Xia H, Gao J, et al. (2020) Role of microglia in the dissemination of Zika virus from mother to fetal brain. PLoS Negl Trop Dis 14(7): e0008413. https://doi.org/10.1371/journal.pntd.0008413
Reviewer 3 Report
The review "The role of microglia during West Nile virus infection of the central nervous system" provides a concise overview of the role microglia play in both controlling infection with West Nile virus in the CNS and in contributing to neurological damage. It is difficult to fault the authors although there is some repetition when reaching for bland conclusions such as "further studies " (line 222), "future studies" (line 269) and repeated use of "Understanding". However, this should not detract from what is a very readable review. The following corrections are suggested:
Line 21. perhaps refer to microglia as resident immune cells.
Line 36. Culex pipiens in italic.
Line 45. "introduction" rather than induction.
Line 70. Could the authors define "older age"
Line 89. "danger" does not seem appropriate (Lassie senses danger, maybe not microglia), perhaps infection would be more appropriate.
Line 97. Microglia can be beneficial or detrimental, or as the authors conclude later in the review, both.
Line 105. 300um?
Figure legend. Define SCSC.
Line 154. "Microglia"
Line 167. "TLR3-deficient" for consistency.
Line 208. "microglia play"
Line 267. suggest "we have highlighted"
Line 285. "leucopenic"
Table. Row 7, column 4, delete "to".
